# Open Domain Multi-document Summarization: A Comprehensive Study of Model Brittleness under Retrieval

**John Giorgi**[1,2,3*†]    **Luca Soldaini**[4†]    **Bo Wang**[1,3]    **Gary Bader**[1,2]
**Kyle Lo**[4†]    **Lucy Lu Wang**[4,5†]    **Arman Cohan**[4,6†]

[1]University of Toronto    [2]Terrence Donnelly Centre    [3]Vector Institute for AI
[4]Allen Institute for AI    [5]University of Washington    [6]Yale University
john.giorgi@utoronto.ca, {lucas, kylel}@allenai.org, lucylw@uw.edu, arman.cohan@yale.edu

## Abstract

Multi-document summarization (MDS) assumes a set of topic-related documents are provided as input. In practice, this document set is not always available; it would need to be retrieved given an information need, i.e. a question or topic statement, a setting we dub "open-domain" MDS. We study this more challenging setting by formalizing the task and bootstrapping it using existing datasets, retrievers and summarizers. Via extensive automatic and human evaluation, we determine: (1) state-of-the-art summarizers suffer large reductions in performance when applied to open-domain MDS, (2) additional training in the open-domain setting can reduce this sensitivity to imperfect retrieval, and (3) summarizers are insensitive to the retrieval of duplicate documents and the order of retrieved documents, but highly sensitive to other errors, like the retrieval of irrelevant documents. Based on our results, we provide practical guidelines to enable future work on open-domain MDS, e.g. how to choose the number of retrieved documents to summarize. Our results suggest that new retrieval and summarization methods and annotated resources for training and evaluation are necessary for further progress in the open-domain setting.[1]

## 1 Introduction

Summarization is an NLP task that aims to generate accurate and coherent summaries for some given text automatically. *Multi-document* summarization (MDS) extends this task to provide multiple topic-related documents as input, with the goal of summarizing salient information while avoiding redundancy. MDS is a popular research objective with many proposed approaches (Yasunaga et al., 2017; Liao et al., 2018; Liu and Lapata, 2019; Li et al., 2020; Jin et al., 2020; Mao et al., 2020; Zhang et al., 2020a; Pasunuru et al., 2021b; Xiao et al., 2022)

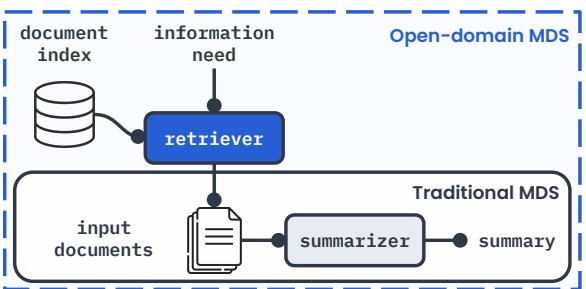

Figure 1: "Traditional" MDS assumes a topic-related set of documents is given at train and test time. Here, we investigate the more challenging "open-domain" setting, where the document set must be retrieved given a query.

and important applications, e.g. the summarization of news articles (Fabbri et al., 2019; Gholipour Ghalandari et al., 2020), scientific literature (Lu et al., 2020; Wallace et al., 2021; DeYoung et al., 2021), and legal documents (Shen et al., 2022).

Existing MDS task definitions, including query-focused MDS (see §3 for detailed comparison), assume a ground-truth, topic-related document set is provided at train and test time. This document set is often an artifact of the dataset curation process; in many practical settings, it is not available a priori and would need to be defined by an *information need*, expressed as a query.[2] Documents relevant to the query would need to be retrieved from a large collection of (mostly irrelevant) documents and summarized (Figure 1); a setting we refer to as *open-domain* MDS.[3] Because even state-of-the-art information retrieval (IR) methods are imperfect, errors, like the retrieval of irrelevant documents, will occur. It is an open question how existing summarizers behave under this more challenging but realistic setting. Our major contributions are:

- We formalize the task definition of open-domain MDS (§2), and bootstrap its study using existing

---

*Work performed during internship at AI2

†Core contributors. See author contributions

[1]https://github.com/allenai/open-mds

[2]E.g. a question: *"Does vitamin D improve physical capabilities of elderly patients?"* or topic statement: *"Report on vulnerabilities of US power grid & efforts to improve it."*

[3]Inspired by the QA literature, where "open-domain" also denotes the setting where only a query is provided as input

datasets, retrievers and summarizers (§4);

- We evaluate summarizers in the open-domain setting and find, through automated and human evaluations, that performance degrades significantly (§5). Promisingly, we show that additional training in the open-domain setting can reduce this sensitivity to imperfect retrieval (§6);

- Finally, we subject summarizers to an extensive suite of carefully designed input perturbations to determine which retrieval errors are driving the degradation in summarization performance (§7)

Based on our results, we provide detailed, practical guidelines for future work in open-domain MDS. We release all code, model & data artifacts, and experimental results created during our investigation.

## 2 Task Definition: Open-domain MDS

In traditional MDS, a model is given a set of topic-related documents $D = \{d_1, ..., d_k\}$ and must generate a summary $S$ that accurately and coherently summarizes the information in $D$. Such models are typically trained in a supervised fashion to minimize the difference between $S$ and a (usually human-written) reference summary $R$. The goals (and evaluation) remain the same in open-domain MDS, but instead of $D$, the inputs are a query $q$ and a large *index* of documents $\mathcal{D}_{index}$, where $|\mathcal{D}_{index}| \gg |D|$. The specifics of $q$, $\mathcal{D}_{index}$, and $S$ will depend on the application or use-case. For example, in automatic literature review (Wallace et al., 2021; DeYoung et al., 2021), $q$ would be a research question or statement, e.g. *"Is massage therapy effective for people with musculoskeletal disorders?"*, $\mathcal{D}_{index}$ would be a large corpus of scientific literature (e.g. PubMed), and $S$ would be a literature review-style discussion, e.g. *"Massage therapy, as a stand-alone treatment, reduces pain and improves function compared to no treatment..."*

There are multiple ways to approach open-domain MDS. We could think of $q$ as a *prompt* to a large language model (LLM) capable of in-context learning (ICL, Brown et al. 2020); in which case we can view $\mathcal{D}_{index}$ as the LLMs training data and retrieval as happening implicitly during inference. However, because all information is stored in the model's weights, this approach requires extremely large models, cannot produce summaries for events outside the training data, and does not provide provenance for the generated summaries.

Another approach is to introduce an explicit retrieval step over an external knowledge source, which we will refer to as "retrieve-then-summarize." It works as follows: a *retriever* ranks all documents in $\mathcal{D}_{index}$ from most-to-least relevant given $q$. The top-$k$ documents are input to a summarizer; $k$ is a parameter that may be tuned for a particular use case. This approach has some desirable properties: (1) $\mathcal{D}_{index}$ can be updated with new documents without re-training the retriever or summarizer, and (2) it provides provenance for model-generated summaries: the top-$k$ documents. In the remainder of the paper, we focus our investigation on the retrieve-then-summarize approach.

## 3 Related Work

**Query-focused MDS** In query-focused MDS (QMDS, Wang et al., 2013; Feigenblat et al., 2017; Xu and Lapata, 2020; Pasunuru et al., 2021a), a query is provided alongside a set of topic-related input documents and used to guide summarization. For example, *extractive* QMDS methods use query relevance to select the sentences that form the summary. No retrieval from a document index is performed. Here, we propose and investigate the more challenging scenario where, given only a query, the input documents must be retrieved from a large index containing mostly irrelevant documents.

**Previous attempts at open-domain MDS** Liu et al. (2018) proposed the WikiSum dataset. Given the title of a Wikipedia article and a collection of non-Wikipedia reference documents, the goal is to generate the first section of the article. The proposed extractive-abstractive approach resembles open-domain MDS. However, the document index is small (10s to 100s of documents) and composed of relevant documents (references of the article plus ten pages of search results using the title as query). We study the more challenging and more general setting where the index is large (»10,000 documents, see Table 6) and contains many more irrelevant documents than relevant documents.

In Zhang et al. (2021), a method similar to retrieve-then-summarize is proposed, using a pre-trained dense passage retriever (Karpukhin et al., 2020) and T5 (Raffel et al., 2020) as summarizer.[4] The model is trained & evaluated on a dataset constructed from existing QMDS datasets. This dataset is small (∼90 training examples) and does not appear to be publicly available. Here, we conduct a large-scale analysis on multiple datasets from different domains (each consisting of thousands of ex-

---

[4]At the time of writing, this work is unpublished

amples) and evaluate several of the top-performing multi-document summarizers currently available (§4). We also extensively simulate document retrieval errors to probe their relative impact on summarization (§7). Together, this allows us to draw broader conclusions about open-domain MDS and provide detailed practical advice for future work.

**Open-domain QA** Our open-domain MDS proposal mirrors a similar trend in question answering (QA). While earlier research focused on answering questions provided a text passage (Rajpurkar et al., 2016, 2018), the now predominant approach, open-domain QA (ODQA), is to answer questions *without* this passage, usually by referencing an external knowledge source (e.g. Wikipedia). Even broader are knowledge-intensive (KI) language tasks (Petroni et al., 2021), which include ODQA but also, for example, fact-checking. KI tasks are commonly approached with a retrieve-then-*generate* framework (Guu et al., 2020; Lewis et al., 2020b; Borgeaud et al., 2022). Retrieve-then-*summarize* is similar, except that the outputs are, on average, much longer and tend to be less extractive than the outputs of KI language tasks like ODQA.

## 4 Bootstrapping Open-domain MDS

Since no large-scale annotated datasets[5] or trained models exist for open-domain MDS, we bootstrap this task using existing datasets (§4.1) and models (§4.2, §4.3). We describe operationalization considerations in §4.4 and evaluation in §4.5.

### 4.1 Datasets

We investigate a representative selection of 5 MDS datasets comprised of news articles, medical studies, and scientific literature, deliberately choosing datasets for which high-performing summarizers exist (see Appendix A for more details). The inputs of these datasets generally consist only of the documents to summarize. However, Multi-XScience and MSˆ2 each provide additional text as input: the target article's abstract and the target review's background section. In our experiments, we always provide this additional text and do not retrieve it.

### 4.2 Retrieval Models

Broadly speaking, retrievers are divided into two categories, *sparse & dense*. Sparse retrievers determine relevance of a document to a query using weighted counts of overlapping terms. Dense retrievers embed documents & queries into a shared vector space and use proximity to determine relevance. Retrievers from these families exhibit different characteristics and limitations (MacAvaney et al., 2022); therefore, we investigate a representative retriever from each: BM25 (sparse, Robertson et al., 1994) and Contriever (dense, Izacard et al., 2022). Both achieve strong zero-shot performance,[6] making them particularly suitable for our purposes. See Appendix B for details.

### 4.3 Multi-document Summarization Models

All MDS models we experiment with are transformer-based encoder-decoders (Vaswani et al., 2017) trained for abstractive summarization, representing the state-of-the-art approach. The input contains one or more documents concatenated with special tokens (e.g. `<doc-sep>`). As is typical, we truncate each document based on the maximum input length of the model divided by the total number of documents. See Appendix C for details.

### 4.4 Operationalize Retrieve-then-Summarize

To extend these datasets and models to the open-domain setting and operationalize the retrieve-then-summarize approach, we address the following:

**How to choose a query?** In open-domain MDS, a query is anything that defines the documents to summarize, e.g. a question or topic statement. Ideally, a human-written query would be available for each example in our dataset; however, existing MDS datasets do not provide these queries. Therefore, we use $R$,[7] the human-written reference summaries, as *pseudo*-queries, as they naturally describe the input documents of each example.[8]

**How to assemble the document index?** For our purposes, we take the set of all documents in the train, validation, and test splits of each dataset to form $\mathcal{D}_{\text{index}}$. This guarantees that the ground-truth documents for each example are present in the index while providing plenty of negative examples.

**How many documents to summarize?** The number of retrieved documents to summarize, $k$,

---

[5]Existing query-focused MDS datasets (e.g. DUC 2005, 2006 & 2007) are extremely small (10s of examples) and are therefore not suitable for the large-scale analysis we conducted

[6]See the BEIR (Thakur et al., 2021) zero-shot benchmark
[7]Except for MSˆ2, where we found the provided "background" section to perform better as a query; see Appendix A
[8]We also experimented with query generation using LLMs (e.g. GPT-3), but found that they significantly underperformed the reference summary as query, e.g. by at least 8 points P/R@K on a sample of the Multi-News validation set

Table 1: Results of the open-domain MDS experiments. We observe the following: (1) retrieval performance ranges from high (Multi-News, WCEP-10, **dark blue**) to low (Multi-XScience, MSˆ2, Cochrane), (2) when summarizers trained on these datasets are provided retrieved documents, they suffer from significant drops in performance (**dark red**); more severe performance drops were observed in cases where baseline summarization performance was relatively high (**dark green**). Experiments here used a sparse retriever (BM25) and *max* top-$k$ strategy (see Table 9 for complete results with all top-$k$ strategies). Similar results were observed using a dense retriever (Contriever, see Table 10). Statistically significant results are underlined (paired t-test, p = 0.01).

| Dataset | Model | Retrieval | | Summarization | | | |
|---|---|---|---|---|---|---|---|
| | | P@K | R@K | ROUGE-Avg F1 | Δ ROUGE-Avg F1 | BERTScore F1 | Δ BERTScore F1 |
| Multi-News | PRIMERA | 0.22 | 0.82 | 31.66 | -7.39 | 31.78 | -10.33 |
| | PEGASUS | – | – | 31.23 | -8.49 | 29.88 | -10.87 |
| | LSG-BART-base | – | – | 30.05 | -6.44 | 26.57 | -8.17 |
| | GPT-3.5-turbo | – | – | 23.86 | -2.46 | 21.68 | -3.92 |
| WCEP-10 | PRIMERA | 0.63 | 0.67 | 35.50 | -1.02 | 48.26 | -0.76 |
| | LSG-BART-base | – | – | 35.76 | -1.15 | 48.17 | -0.85 |
| | GPT-3.5-turbo | – | – | 26.36 | -0.22 | 32.72 | -0.25 |
| Multi-XScience | PRIMERA | 0.06 | 0.40 | 18.31 | -0.57 | 10.57 | -1.82 |
| MSˆ2 | LED-base | 0.16 | 0.22 | 19.66 | -0.14 | 22.74 | -0.47 |
| Cochrane | LED-base | 0.17 | 0.57 | 17.39 | -0.28 | 23.12 | -2.11 |

is a parameter that can be tuned for different use cases. To determine its impact on summarization performance, we investigate three strategies:

- **Max**: Choose $k$ as the *maximum* number of input documents for any example in a given dataset. Tends to select for *recall* at the cost of *precision*.

- **Mean**: Choose $k$ as the *mean* number of input documents for all examples in a given dataset. Tends to select for *precision* at the cost of *recall*.

- **Oracle**: Choose $k$ as the number of *ground-truth* input documents for each example. This mimics the scenario where all documents with a relevance score (assigned by the retriever) above a certain optimal threshold are retained.

We note that this is a highly idealized setting. Using $R$ as query leaks information about the reference summary into the retrieval step, likely inflating retrieval and summarization performance. In practice, $\mathcal{D}_{index}$ will be much larger (e.g. PubMed-, Wikipedia-, or even Web-scale), making retrieval more difficult. Our intention is to determine a *lower-bound* for the expected performance degradation of state-of-the-art summarizers in the open-domain setting; as we will show in §5, even this idealized setting often leads to large reductions in performance. In §7, we extensively *simulate* document retrieval errors to determine how summarizers behave in both low- and high-performing retrieval settings across a variety of retrieval error types.

### 4.5 Evaluation

We follow previous work by evaluating summarization with ROUGE-1/2/L scores (Lin, 2004). To provide a single metric for comparison, we report

ROUGE-Avg F1, the average F1-score of ROUGE-1/2/L. We also report BERTScore (Zhang et al., 2020b), which has been shown to better correlate with human judgment (Yuan et al., 2021; Fischer et al., 2022). For document retrieval, we report the precision and recall at $k$ (P@K and R@K); which are suitable metrics when the input documents do not have an inherent order, as is usually the case in MDS. We evaluate on the test splits of each dataset, except for MSˆ2 and Cochrane, where we evaluate on the validation set because the test split is blind.

## 5 Evaluating Open-domain MDS

Here we present the results of our open-domain MDS experiments. In general, we find existing summarizers suffer large reductions in performance when applied to open-domain MDS, even when retrieval performance is high (Table 1).[9] Below, we provide key observations on how the individual components (retriever and summarizer) behave within a retrieve-then-summarize framework.

**Strong summarizers more sensitive to imperfect retrieval than weak ones** We observe a relationship between a summarizer's (baseline) performance on a dataset and its sensitivity to imperfect document retrieval (Table 1). The largest reductions in summarization performance were observed for the most performant summarizers, despite retrieval performance being the highest in these cases. However, this relationship is confounded by differences between datasets. To control for this, we conduct experiments comparing fine-tuned PRIMERA

---

[9]Results for sparse and dense retrievers were comparable and exhibited similar trends. We elect to show results for the sparse retriever; see Appendix E for dense retriever results

Table 2: Results of the open-domain MDS experiments with zero-shot summarizers. Controls for differences in datasets and models, isolating the relationship between summarization performance in the traditional and open-domain settings. Top-$k$ strategy *mean* is used. Statistically significant results are underlined (paired t-test, p = 0.01).

| Dataset | Model | Retrieval | | | Summarization | | | |
|---|---|---|---|---|---|---|---|---|
| | | Retriever | P@K | R@K | ROUGE-Avg F1 | Δ ROUGE-Avg F1 | BERTScore F1 | Δ BERTScore F1 |
| Multi-News | PRIMERA | sparse (BM25) | 0.64 | 0.74 | 31.66 | -2.82 | 31.78 | -4.08 |
| | | dense (Contriever) | 0.59 | 0.70 | – | -3.31 | – | -4.60 |
| | ↪ *zero-shot* | sparse | – | – | 23.58 | -0.09 | 18.66 | -0.39 |
| | | dense | – | – | – | -0.27 | – | -0.44 |
| WCEP-10 | PRIMERA | sparse | 0.66 | 0.64 | 35.50 | -0.90 | 48.26 | -0.68 |
| | | dense | 0.66 | 0.63 | – | -0.14 | – | +0.68 |
| | ↪ *zero-shot* | sparse | – | – | 21.43 | +0.35 | 25.48 | +0.72 |
| | | dense | – | – | – | +1.00 | – | +2.19 |
| Multi-XScience | PRIMERA | sparse | 0.16 | 0.27 | 18.31 | -0.25 | 10.57 | -1.27 |
| | | dense | 0.16 | 0.24 | – | -0.81 | – | -0.96 |
| | ↪ *zero-shot* | sparse | – | – | 15.18 | +0.69 | 6.02 | -0.47 |
| | | dense | – | – | – | +0.46 | – | +0.00 |

Table 3: Comparing ROUGE-Avg F1 scores of model-generated summaries to heuristic baselines. In some cases, heuristics perform surprisingly close to trained summarizers. All Lead is the concatenation of the first sentence from each input document. Oracle document is the document with the highest token overlap with the reference summary; oracle lead is the first sentence from this document. Background/Abstract is the additional input from MSˆ2 and Multi-XScience. The best baseline for each dataset is **bolded**.

| Dataset | Best Summarizer | Baselines | | | | |
|---|---|---|---|---|---|---|
| | | Δ Random Summary | Δ All Lead | Δ Oracle Document | Δ Oracle Lead | Δ Background/Abstract |
| Multi-News | 31.7 | -18.3 | -15.3 | **-4.1** | -21.8 | – |
| WCEP-10 | 35.8 | -27.8 | -24.4 | -15.3 | **-9.9** | – |
| Multi-XScience | 18.3 | -6.2 | -5.0 | **-0.8** | -9.3 | -2.3 |
| MSˆ2 | 19.7 | -10.4 | -11.0 | -7.6 | -4.0 | **-0.2** |
| Cochrane | 17.4 | -5.0 | -4.2 | -3.9 | **-3.5** | – |

to PRIMERA evaluated *zero-shot* (Table 2).[10] This allows us to hold the dataset, model architecture, and retriever constant, isolating the relationship between summarization performance in the traditional and open-domain settings. Here, the trend is clear: "strong" summarizers are more sensitive to imperfect retrieval than "weak" summarizers.[11]

One explanation is that weak summarizers have less to lose from imperfect retrieval, perhaps because they are not performing adequately even when trained and evaluated on ground-truth inputs. They may, to a greater degree than strong summarizers: hallucinate, exploit heuristics, or use only a fraction of the input documents (Kryscinski et al., 2019; Wolhandler et al., 2022). To probe this, we construct several baselines that mimic these behaviours. We find that, for example, copying the background section of MSˆ2 performs comparably to a trained model, suggesting that the observed insensitivity to retrieval errors could be due to summarizers exploiting this heuristic (Table 3, see Appendix E.1 for details). We observe a similar result

for Multi-XScience by copying the document with the highest token overlap to the reference summary. Future work should carefully establish that summarizers are performing adequately before attempting the more difficult open-domain setting.

**Better retrieval performance ≠ better summarization performance** Performance of the sparse and dense retrievers was generally comparable (Table 7), with the sparse retriever performing better on some datasets (Multi-News, WCEP-10, Multi-XScience, see Table 9) and the dense retriever performing better on others (MSˆ2, Cochrane, see Table 10). Interestingly, however, better retrieval performance did not always correspond with smaller reductions in summarization performance. E.g. on WCEP-10, the sparse retriever performed slightly better, but the reduction in summarization performance was considerably larger. On MSˆ2 and Cochrane, the better-performing dense retriever led to a larger reduction in summarization performance. This suggests that the two types of retrievers are making characteristically different errors[12] that P/R@K do not completely capture. Future work should consider summarization performance

---

[10]The only model we evaluate with zero-shot capabilities

[11]We use "strong" and "weak" as shorthand to refer to cases where summarization performance is high (e.g. PRIMERA on Multi-News) and low (e.g. LED on Cochrane)

[12]Previously noted by MacAvaney et al. (2022)

Table 4: Human evaluation on Multi-News. A binomial test on three human annotators for $n = 50$ random test examples was conducted for each facet (excluding ties). All results statistically significant ($p < 0.01$). Inter-annotator agreement reported as Fleiss' Kappa ($\kappa$).

| Facet | baseline preferred | open-domain preferred | p | $\kappa$ |
|---|---|---|---|---|
| Coverage | 60 | 23 | 5.97e-05 | 0.32 |
| Informativeness | 69 | 27 | 2.15e-05 | 0.47 |

itself, *alongside* IR metrics like P/R@K, when tuning retrieval pipelines for open-domain MDS.

**The number of documents to retrieve matters** We observe clear differences in the strategy for choosing $k$, the number of retrieved documents to summarize. Unsurprisingly, the oracle strategy almost always leads to the smallest reduction in summarization performance. This strategy closely mimics the setting of retaining all documents with a relevance score (assigned by the retriever) over a certain threshold but assumes a strong retriever and a well-calibrated threshold, both of which may be difficult to achieve in practice. Our results suggest that setting $k$ as the *mean* number of relevant documents (if an accurate estimate can be made) is a second-best strategy. We note that, relative to max $k$, mean $k$ tends to select for precision over recall (see P@K vs. R@K scores in Table 1 & Table 10); future work should consider tuning $k$ for precision.

**Human evaluation confirms degradation of summarization performance** Automatic evaluation metrics like ROUGE are imperfect and may not correlate with aspects of human judgment.[13] Therefore, we conducted a human evaluation to validate our results. In short, human annotators have a statistically significant preference for summaries produced by the "baseline" model (no retrieval) along two facets, *coverage* and *informativeness* (Table 4), corroborating the degradation of summarization performance in the open-domain setting as quantified by the automatic metrics. See Appendix H for full details, including a manual analysis of example summaries produced in the open-domain setting.

**In-Context Learning with LLMs** In-context learning (ICL) with large language models (LLMs) has emerged as a viable approach to zero-shot summarization (Goyal et al., 2022). We conducted preliminary experiments using this approach to determine how its behaviour in the open-domain set-

[13]See §8 for an extended discussion

**Prompt Template**

**Natural language instructions**
You are an expert journalist. Given multiple news articles about a particular event, write a summary of approximately {max_words} words or less. Respond in "journalese". Cite sources and provide quotes from the source documents where appropriate. Do not refuse to answer. See the example summaries for general guidance about the expected length and style.

**In-context examples (up to 5)**
Example summaries
{examples}

**Test input**
Source documents
{documents}
Summary:

Figure 2: Prompt template for our in-context learning (ICL) based approach. Each prompt includes natural language instructions, example reference summaries as in-context examples, and unseen source documents as input. max_words is set per dataset, 384 for Multi-News and 32 for WCEP-10. examples (2 for Multi-News and 5 for WCEP-10) are randomly selected from the train set, and the same examples are used for every input.

ting compares to the fine-tuned models that were the focus of our evaluation. We chose GPT-3.5-turbo as the LLM and designed a suitable prompt (tuned based on validation set performance) which contains some natural language instructions and a few example summaries (Figure 2). We omit experiments on the MS^2 and Cochrane datasets, whose source documents, even when truncated to the first 25, exceed GPT-3.5's maximum input token length of 4096 (see Table 6). We also omit experiments on Multi-XScience, as a snapshot of arXiv is presumably included in the model's training set; therefore we cannot control for possible train-test contamination as the reference summaries are the related work sections of arXiv papers. To fit within the maximum token limit of the model, we use only example summaries as the in-context examples (omitting the source documents) and randomly choose 2 example summaries from the train-set for Multi-News and 5 for WCEP-10. To make results as reproducible as possible, we set the temperature=0 and used the 03/01/2023 GPT-3.5-turbo snapshot. All other hyperparameters of the OpenAI API are left at their defaults.[14] Lastly, due to associated costs in using the model, we restrict our experi-

[14]https://platform.openai.com/docs/api-reference/completions

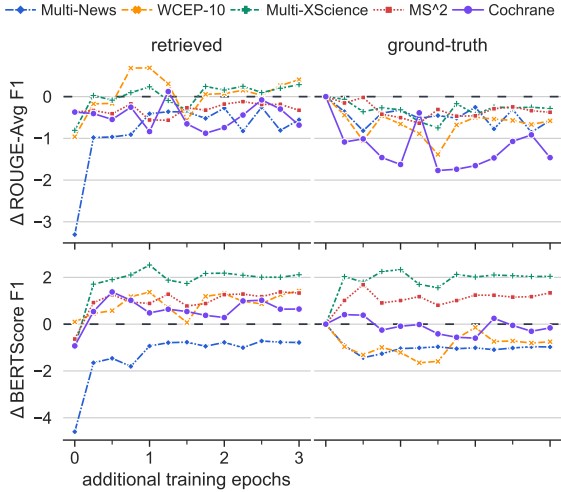

Figure 3: Fine-tuning summarizers in the open-domain. Additional training on *retrieved* documents can reduce sensitivity to imperfect retrieval; often at the cost of performance on *ground-truth* documents. Dashed **grey** line represents no change in performance.

ments to the sparse retriever. Results are presented in Table 1. The ICL-based approach significantly underperforms the fine-tuned models (by at least ~8 points) but outperforms PRIMERA *zero-shot*. The general trends (and relative magnitude) of the degradation in performance in the open domain are comparable. We note that previous work has found that human annotators often prefer GPT-generated summaries over those generated by smaller, fine-tuned models, even when automatic metrics like ROUGE disagree. However, this observation is restricted to the single-document setting (Goyal et al., 2022). Future work is needed to determine if LLMs exhibit different behaviour under the open-domain MDS setting than their smaller, fine-tuned pre-trained-language model (PLM) counterparts.

## 6 Training in the Open-domain Setting

A natural question is whether a summarizer's robustness to document retrieval errors at *test* time can be improved by exposing the model to similar errors at *train* time. To explore this, we retrieve the documents for all examples in the train set of each dataset and fine-tune summarizers on these examples. We then evaluate them on both the retrieved document set and the original (ground-truth) document set (Figure 3, see Appendix F for details). We find clear cases where summarization performance in the open-domain benefits from the additional training (e.g. Multi-News, Multi-XScience), but that this benefit can come at the cost of performance on the ground-truth documents (e.g. Multi-News,

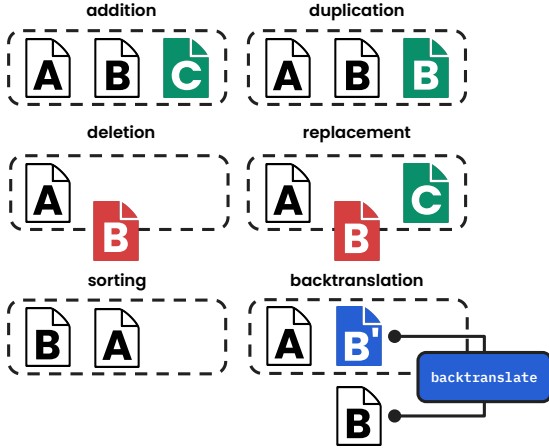

Figure 4: Graphical depiction of perturbations. Dashed line indicates input documents. Uncoloured documents are ground-truth, **green** have been added, **red** removed and **blue** modified. Unique documents are lettered.

WCEP-10 and Cochrane), and is sometimes unstable (e.g. WCEP, Cochrane). We note again that our retrieval setting is highly idealized (§4.4); nonetheless, our results suggest that existing summarizers could be adapted to the open-domain setting if query-annotated training examples and appropriate document indices are constructed.

## 7 Simulating Document Retrieval Errors

In this section, we investigate what is driving the reduction in summarization performance in the open-domain setting (§5). We begin by carefully categorizing the various retrieval errors that can occur. E.g., we could erroneously retrieve documents *irrelevant* to the query. For each error type, we design a corresponding "perturbation" that can be applied to the inputs of existing MDS datasets before they are fed to a summarizer. The perturbations are described below and depicted graphically in Figure 4:

- **Addition**: *Add one or more irrelevant documents*. This could occur if we retrieve all relevant documents but also retrieve irrelevant ones.

- **Deletion**: *Remove one or more documents*. This could occur if we retrieve only a fraction of all relevant documents.

- **Replacement**: *Replace one or more relevant documents with irrelevant ones*. This could occur if we retrieve the correct number of documents but substitute relevant ones for irrelevant ones.

- **Duplication**: *Duplicate one or more documents*. This could occur if duplicate (or, more likely, near-duplicate) documents exist in the index.[15]

---

[15]Deduplication is non-trivial (Lee et al., 2022) and near-

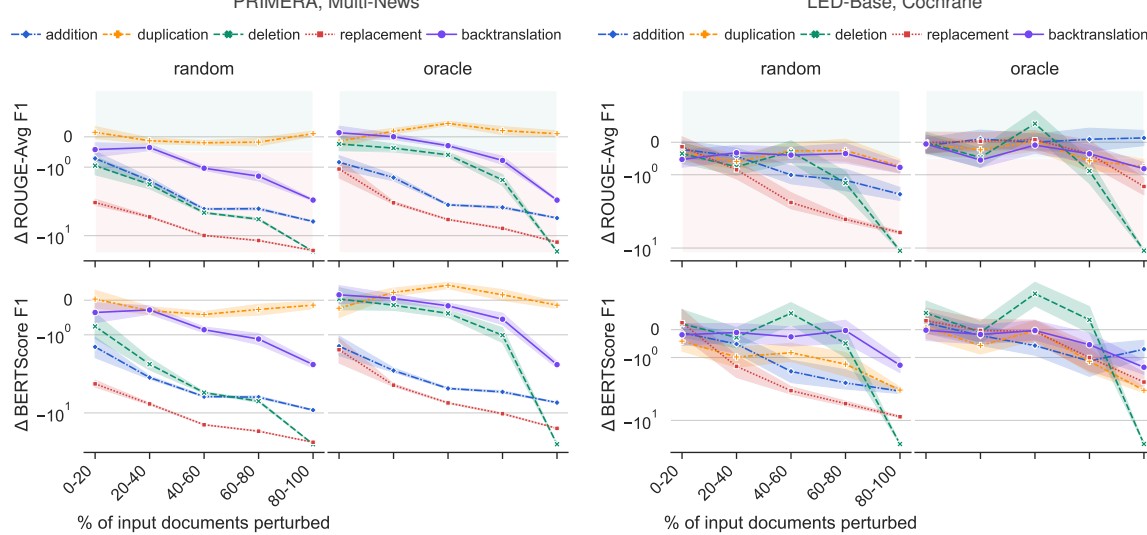

Figure 5: Results of the perturbation experiments on Multi-News (left) and Cochrane (right). Mean change in summarization performance plotted against the percent of perturbed input documents. Values above -0.49 ROUGE are shaded in **green**, and values below in **red**, the average difference in summarization performance reported in *CL conferences. Y-axis is displayed in symlog scale. 68% confidence intervals (CI) are plotted as error bands.

- **Sorting**: *Modify order of documents*. Input documents for MDS are typically unordered. However, many methods concatenate documents before providing them as input, and it is unknown if models are sensitive to this order. Different orderings could occur, e.g., if documents are sorted by order of relevance before concatenating.

**Token-level perturbation**   It is well known that NLP models are sensitive to minor *token*-level changes in their inputs (Prabhakaran et al., 2019; Niu et al., 2020; Ribeiro et al., 2020; Moradi and Samwald, 2021). To compare and contrast the *document*-level sensitivity we investigate with this known sensitivity, we include a token-level perturbation, **backtranslation**, as control. Here, we translate one or more input documents to another high-resource language and back again. This causes small changes, e.g. paraphrasing and synonym substitution, allowing us to create semantics-preserving, token-level changes to a document.[16]

### 7.1   Selecting Documents to Perturb

Each perturbation requires selecting one or more documents to perturb, e.g., in addition and deletion, we need to choose which documents to add and which to remove. We investigate two strategies:

- **Random**: Select documents to perturb randomly, mimicking a (very) *weak* retriever.

- **Oracle**: Select documents in a way that mimics a *strong* retriever. E.g. in deletion, we remove ground-truth documents in order of *least-to-most* similar to the reference summary $R$,[17] in addition, we add irrelevant documents in order of *most-to-least* similar.[18]

For perturbations that require selecting irrelevant documents (addition & replacement), we select from the set of all documents in the train, validation, and test splits (excluding documents from the example we are perturbing). We evaluate summarizers under increasing amounts of perturbation: from 0% of documents perturbed up to 100%.

### 7.2   Results of Simulation Experiments

In Figure 5, we display the results of our experiments simulating document retrieval errors for two model-dataset pairs.[19] To better contextualize results, we shade differences in ROUGE $\geq 0.5$ (average difference in summarization performance reported in *CL conferences, Deutsch et al., 2022) in **red** and the rest in **green**. This serves as a rough yardstick to help identify large drops in performance. We symlog (Webber, 2012) the y-axis to make small changes in performance more apparent. In general, results are congruent with our open-domain MDS experiments (§5): (1) large reduc-

---

duplicates are not uncommon in large document collections like C4 (Dodge et al., 2021) or S2ORC (Lo et al., 2020)

[16]See Appendix G.1 for details

[17]Similar to §5, this leverages $R$ as a pseudo-query

[18]Determined using Sentence Transformers (Reimers and Gurevych, 2019); specifically all-MiniLM-L6-v2

[19]These are exemplary of main trends observed across all model-dataset pairs; see Appendix G for complete results

tions in summarization performance are observed even in cases of few simulated errors (< 20%) and (2) strong summarizers (Figure 5, left) are more sensitive to retrieval errors than weak summarizers (Figure 5, right). Below, we discuss major trends.

**Summarizers insensitive to duplicates and small token-level changes**  A consistent trend across models & datasets was an insensitivity to duplicate documents, even in the extreme case of >80% duplication, suggesting that deduplication efforts on the document index are unlikely to translate to improvements in summarization performance. However, this assumes duplicate documents are included without replacing relevant documents, which is possible if $k$ is based on a relevance threshold cutoff. Another trend is that models are not overly sensitive to minor *token*-level changes (exemplified by backtranslation) relative to other perturbations, further motivating our focus on *document*-level errors.

**Erroneous additions vs. erroneous deletions**  In the *random* setting, small amounts of deletion led to large drops in summarization performance. Conversely, deletion has surprisingly little impact in the *oracle* setting until most documents (>60%) are removed. These results have two non-mutually exclusive explanations: (1) summarizers only consider a select few input documents, (2) reference summaries have low coverage of the input documents, both corroborated by recent work (Wolhandler et al., 2022). Based on the addition perturbation results in the oracle setting, summarizers appear to be more sensitive to erroneous *additions* than *deletions*, which, alongside our top-$k$ strategy results in §5, suggests that retrieval pipelines should be tuned for *precision* in open-domain MDS.

**Summarizers are insensitive to document order** As far as we know, prior work has yet to investigate whether multi-document summarizers are sensitive to input document order. Although the documents are generally considered unordered, they are usually concatenated before providing them as input. To determine if summarizers are sensitive to this order, we sorted the input documents of each dataset *before* concatenation and re-evaluated the summarizers. We investigate two sorting strategies:

- **Random**: Shuffle documents randomly.

- **Oracle**: Sort documents by similarity to the reference summary, $R$. This is motivated from two perspectives: (1) prior work has found that trans-

formers are biased toward earlier tokens in their input (Hofstätter et al., 2021), so we might expect improved performance by placing the most similar content to $R$ first, (2) a strong retriever would assign a higher rank to the most relevant documents, and we might choose to input documents to our summarizer in this order.

We find no significant difference (paired t-test, p = 0.01) in summarization performance for any model-dataset pair, *except* for WCEP-10 (see Appendix G.2). Here we find that both models we evaluate (PRIMERA & LSG-BART) are negatively affected by random sorting. One possible explanation is that, due to how WCEP-10 was constructed, the documents are (partially) sorted in order of relevance (see Appendix A). Models trained on this dataset may have learned to exploit this, e.g., by assigning more weight to earlier documents in the input. After randomly shuffling input documents, this learned heuristic would no longer hold, and summarization performance might drop accordingly.

## 8   Conclusion

We present a new, open-domain task definition for MDS. This reformulation is more challenging and potentially more useful, enabling users to specify their intent with only a query. Via extensive automatic and manual evaluation, we find that: (1) summarization performance significantly degrades in the open-domain setting, even when retrieval performance is high, (2) additional training can reduce this sensitivity to imperfect retrieval, and (3) summarizers are insensitive to the retrieval of duplicate documents and the order of retrieved documents but highly sensitive to other errors, like the retrieval of irrelevant documents. Based on our results, we provide practical guidelines, e.g. that retrieval pipelines for open-domain MDS should be tuned for *precision*. Curating high-quality MDS datasets annotated with queries will be necessary to enable further progress in the open-domain setting.

## Limitations

**Automated evaluation metrics may not correlate with human judgment**  Though established metrics such as ROUGE and BERTScore are imperfect (Deutsch et al., 2022), they are frequently used in the summarization literature, do correlate with aspects of summary quality, and are useful for comparing system-level performance, especially in scenarios such as ours where performance differ-

ences can be several points below the baseline. To validate our findings, we also conduct a human evaluation to better understand qualitative differences in summaries generated in the open-domain setting (see Appendix H). The investigation of better automated metrics for natural language generation is an active field of research, and we hope to integrate novel and performant metrics in future work.

**Results conflate dataset features and model performance**  Our evaluation conflates several issues beyond the relative performance of retrievers and summarizers. Dataset quality, the "multi-document-ness" of each dataset, and the shortcomings of automatic metrics all contribute to noise in our results. For example, a dataset whose reference summaries have low coverage of the input documents (as characterized by Wolhandler et al., 2022) would not be expected to respond to retrieval errors in the same way as a dataset where this coverage is high. By experimenting with multiple datasets, retrievers, and summarizers, as well as in the synthetic perturbation setting (§7), we expect our results to be more resilient to these confounders.

**Specialized retrievers may lead to better performance**  We experiment with standard sparse and dense retrievers in the zero-shot setting. More effort tuning retrieval pipelines, e.g. by introducing re-rankers (Pradeep et al., 2021) or by fine-tuning retrievers directly on MDS datasets, may improve retrieval performance and lead to smaller summarization performance reductions. Additionally, better summarization performance might be achieved by retrieving content at the *span*-level, (as opposed to full documents). We leave the development of retrieval pipelines purpose-built and tuned for open-domain MDS to future work.

## Acknowledgements

This research was enabled in part by support provided by the Digital Research Alliance of Canada (alliancecan.ca) and Compute Ontario (www.computeontario.ca).

## Author Contributions

John Giorgi made most of the technical contributions, including dataset collection and processing, model implementation, and running experiments. John also contributed to project scoping and ideation, wrote the paper with feedback from everyone, and led the project in general. Luca, Kyle, Lucy, and Arman were project mentors, contributing equally to project scoping and experimental design and providing the core ideas and direction throughout the course of the project and paper writing. Additionally, Luca made technical contributions to model implementation. Bo and Gary provided high-level feedback and advice in later stages of the project.

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

## A   Dataset Details

All datasets were managed in the HuggingFace Datasets library (Lhoest et al., 2021). The examples of each dataset consist of an input document set, $D$ and a human-written reference summary, $S$. Multi-XScience and MS^2 each have an additional input that is always provided (and never retrieved or perturbed), the target articles abstract and the target reviews background section. See Table 6 for dataset statistics. Below, we provide detailed descriptions of each dataset:

- **Multi-News** (Fabbri et al., 2019): Consists of news articles and summaries collected from www.newser.com. There are 44,972 examples in the train set and 5622 examples in the test set. Each example contains between 1 and 10 documents, with a mean of ∼2.7.

- **WCEP-10**: Consists of news articles and summaries collected from the Wikipedia Current Events Portal (WCEP[20]). WCEP-10[21] sub-samples the top 10 most relevant documents from the original WCEP dataset (Gholipour Ghalandari et al., 2020). There are 8158 examples in the train set and 1022 examples in the test set. Each example contains between 1 and 10 documents, with a mean of ∼9.1.

- **Multi-XScience** (Lu et al., 2020): The target summary of each example is the related works section of a scientific article, and the input documents are the abstracts of the articles this section cites. Also included is the target article's abstract. There are 30,369 examples in the train set and 5093 examples in the test set. Each example contains between 1 and 20 documents, with a mean of ∼4.1.

- **MS^2** (DeYoung et al., 2021): The target summary is a few sentences from a biomedical systemic review which summarize the main findings. The input documents are the included studies for that review. Also included is the target reviews background section. There are 14,188 examples in the train set and 2021 examples in the validation set. Each example contains between 1 and 401 documents, with a mean of ∼23.2.

Table 5: Evaluated multi-document summarizers and the datasets for which a fine-tuned model is publicly available (or was trained by us).

| Model | Fine-tuned on | Max Input Len. | Zero-shot? |
|---|---|---|---|
| LED | MS^2, Cochrane | 16384 | ✗ |
| PEGASUS | Multi-News | 1024 | ✗ |
| PRIMERA | Multi-News, WCEP-10, Multi-XScience | 4096 | ✓ |
| LSG-BART | Multi-News, WCEP-10 | 4096 | ✗ |

- **Cochrane** (Wallace et al., 2021): Similar to MS^2, except a background statement is not included as input. There are 3752 examples in the train set and 470 examples in the validation set. Each example contains between 1 and 537 documents, with a mean of ∼10.9.

## B   Retrieval Details

Document retrieval and evaluation are conducted in the PyTerrier library (Macdonald and Tonellotto, 2020). In Table 7, we present the retrieval performance on the train, validation and test split for each dataset, retriever, and top-$k$ strategy. Below, we provided detailed descriptions of all retrievers:

- **BM25** (Robertson et al., 1994): Like other sparse retrievers, BM25 represents queries and documents as sparse vectors, where each element of a vector corresponds to a term in the vocabulary. BM25 is a widely used weighting scheme that extends TF-IDF (Jones, 2004) to account for document length and term-frequency saturation. We use BM25 via PyTerrier with the default settings.

- **Contriever** (Izacard et al., 2022): Contriever is an unsupervised dense retriever that uses a bi-encoder architecture. Documents and queries are encoded independently using the same BERT model (Devlin et al., 2019), and the final embedding is obtained by mean-pooling over the hidden representations of the model's last layer. The relevance score between a query and a document is the dot product of their embeddings. Specifically, we use contriever-msmarco,[22] which has been fine-tuned on the MS MARCO dataset (Campos et al., 2016). We use Contriever via the PyTerrier Sentence Transformers plugin (Soldaini, 2022) with the default settings.

---

[20]https://en.wikipedia.org/wiki/Portal:Current_events

[21]https://huggingface.co/datasets/ccdv/WCEP-10

[22]https://huggingface.co/facebook/contriever-msmarco

Table 6: Dataset statistics, counting whitespace tokens and punctuation. *Following (DeYoung et al., 2021), we take the first 25 documents as input (full statistics in parentheses). †Multi-XScience and MS^2 each have inputs that are always provided (and never retrieved), the target articles abstract and the target reviews background section.

| Dataset | Domain | Number of Documents | | | Average Number of Tokens | |
| --- | --- | --- | --- | --- | --- | --- |
| | | Max | Mean | Total | Per document | Per summary |
| Multi-News | News Articles | 10 | 2.7 | 154,544 | 788 | 267 |
| WCEP-10 | News Articles | 10 | 9.1 | 92,560 | 494 | 33 |
| Multi-XScience† | Scientific Literature | 20 | 4.1 | 165,546 | 153 | 125 |
| MS^2*† | Medical Studies | 25 (401) | 17 (23) | 415,333 | 332 | 58 |
| Cochrane* | Medical Studies | 25 (537) | 9 (11) | 51,208 | 266 | 69 |

Table 7: Retrieval performance. The precision and recall at $k$ for each retriever and top-$k$ strategy is reported. The index for each dataset is the set of all documents in the train, validation and test sets; the reference summaries are used as queries, except for MS^2, where we use the provided background section. The Cochrane test set is blind, so we do not have access to the reference summaries to use as queries and therefore do not evaluate on the test set.

| Dataset | Retriever | Retriever Type | Top-$k$ Strategy | Train | | Validation | | Test | |
| --- | --- | --- | --- | --- | --- | --- | --- | --- | --- |
| | | | | P@K | R@K | P@K | R@K | P@K | R@K |
| Multi-News | BM25 | sparse | max(10) | 0.22 | 0.83 | 0.22 | 0.82 | 0.22 | 0.82 |
| | | | mean (3) | 0.64 | 0.74 | 0.64 | 0.74 | 0.64 | 0.74 |
| | | | oracle | 0.75 | 0.75 | 0.75 | 0.75 | 0.75 | 0.75 |
| | Contriever | dense | max | 0.21 | 0.80 | 0.21 | 0.79 | 0.21 | 0.80 |
| | | | mean | 0.59 | 0.69 | 0.59 | 0.69 | 0.59 | 0.70 |
| | | | oracle | 0.69 | 0.69 | 0.69 | 0.69 | 0.69 | 0.69 |
| WCEP-10 | BM25 | sparse | max (10) | 0.59 | 0.66 | 0.60 | 0.63 | 0.63 | 0.67 |
| | | | mean (9) | 0.62 | 0.62 | 0.63 | 0.60 | 0.66 | 0.64 |
| | | | oracle | 0.64 | 0.64 | 0.63 | 0.63 | 0.67 | 0.67 |
| | Contriever | dense | max | 0.60 | 0.66 | 0.60 | 0.64 | 0.63 | 0.67 |
| | | | mean | 0.62 | 0.63 | 0.63 | 0.60 | 0.66 | 0.63 |
| | | | oracle | 0.65 | 0.65 | 0.63 | 0.63 | 0.66 | 0.66 |
| Multi-XScience | BM25 | sparse | max (20) | 0.05 | 0.41 | 0.06 | 0.40 | 0.06 | 0.40 |
| | | | mean (4) | 0.16 | 0.27 | 0.16 | 0.26 | 0.16 | 0.27 |
| | | | oracle | 0.22 | 0.22 | 0.22 | 0.22 | 0.23 | 0.23 |
| | Contriever | dense | max | 0.06 | 0.38 | 0.06 | 0.38 | 0.06 | 0.38 |
| | | | mean | 0.16 | 0.24 | 0.16 | 0.24 | 0.16 | 0.24 |
| | | | oracle | 0.20 | 0.20 | 0.20 | 0.20 | 0.21 | 0.21 |
| MS^2 | BM25 | sparse | max (25) | 0.17 | 0.26 | 0.16 | 0.22 | 0.17 | 0.22 |
| | | | mean (17) | 0.21 | 0.22 | 0.18 | 0.18 | 0.20 | 0.18 |
| | | | oracle | 0.22 | 0.22 | 0.18 | 0.18 | 0.19 | 0.19 |
| | Contriever | dense | max | 0.19 | 0.29 | 0.18 | 0.25 | 0.19 | 0.26 |
| | | | mean | 0.23 | 0.24 | 0.21 | 0.21 | 0.23 | 0.21 |
| | | | oracle | 0.24 | 0.24 | 0.21 | 0.21 | 0.22 | 0.22 |
| Cochrane | BM25 | sparse | max (25) | 0.17 | 0.55 | 0.17 | 0.57 | – | – |
| | | | mean (9) | 0.30 | 0.42 | 0.31 | 0.44 | – | – |
| | | | oracle | 0.38 | 0.38 | 0.40 | 0.40 | – | – |
| | Contriever | dense | max | 0.20 | 0.63 | 0.20 | 0.64 | – | – |
| | | | mean | 0.34 | 0.48 | 0.35 | 0.49 | – | – |
| | | | oracle | 0.45 | 0.45 | 0.44 | 0.44 | – | – |

## C  Model Details

All models are implemented in PyTorch (Paszke et al., 2019), and pretrained weights are obtained from the HuggingFace Transformers library (Wolf et al., 2020). Models were trained and evaluated on 1-4 NVIDIA A100 40GB GPUs. We list details about the models in Table 5. Below, we provide detailed descriptions of all models:

- **LED** (Beltagy et al., 2020): LED replaces full self-attention with local windowed attention and global attention mechanisms that scale linearly with input sequence length, allowing for efficient processing of inputs up to 16K tokens. Its parameters are initialized with the pretrained parameters of BART (Lewis et al., 2020a), its positional embeddings with 16 copies of BART's 1K position embeddings. The model is fine-tuned on MDS datasets in a supervised fashion.

Table 8: Reported versus reproduced ROUGE-1/2/L scores for each model-dataset pair evaluated in the main paper. We also report zero-shot performance on select datasets for PRIMERA. *Fine-tuned by us.

| Model / Dataset | Reported | | | | Reproduced | | | |
|---|---|---|---|---|---|---|---|---|
| | PRIMERA | PEGASUS | LED-base | LSG-BART-base | PRIMERA | PEGASUS | LED-base | LSG-BART-base |
| Multi-News | 49.9/21.1/25.9 | 47.5/18.7/24.9 | – | 47.1/18.9/25.2 | 49.3/20.3/25.4 | 48.2/20.1/25.4 | – | 46.3/18.8/25.1 |
| ↪ zero-shot | 42.0/13.6/20.8 | – | – | – | 39.7/11.9/19.2 | – | – | – |
| WCEP | 46.1/25.2/37.9 | – | – | 46.0/24.2/37.4 | 45.1/24.7/36.7 | – | – | 45.9/24.1/37.2 |
| ↪ zero-shot | 28.0/10.3/20.9 | – | – | – | 31.3/10.7/22.2 | – | – | – |
| Multi-XScience | 31.9/7.4/18.0 | – | – | – | 31.7/6.1/17.1 | – | – | – |
| ↪ zero-shot | 29.1/4.6/15.7 | – | – | – | 27.0/3.9/14.6 | – | – | – |
| MS^2* | – | – | 26.4/8.0/19.6 | – | – | – | 28.5/9.5/20.9 | – |
| Cochrane* | – | – | 23.9/6.6/17.6 | – | – | – | 26.9/6.9/18.4 | – |

Table 9: Results of the open-domain MDS experiments. We observe the following: (1) retrieval performance ranges from high (Multi-News, WCEP-10, **dark blue**) to low (Multi-XScience, MS^2, Cochrane), (2) when summarizers trained on these datasets are provided retrieved documents, they suffer from significant drops in performance (**dark red**); more severe performance drops were observed in cases where baseline summarization performance was relatively high (**dark green**). Experiments here used a sparse retriever (BM25); similar results were observed using a dense retriever (Contriever, see Table 10). Statistically significant results are underlined (paired t-test, p = 0.01).

| Dataset | Model | Retrieval | | | Summarization | | | |
|---|---|---|---|---|---|---|---|---|
| | | Top-$k$ Strategy | P@K | R@K | ROUGE-Avg F1 | Δ ROUGE-Avg F1 | BERTScore F1 | Δ BERTScore F1 |
| Multi-News | PRIMERA | max (10) | 0.22 | 0.82 | 31.66 | -7.39 | 31.78 | -10.33 |
| | | mean (3) | 0.64 | 0.74 | – | -2.82 | – | -4.08 |
| | | oracle | 0.75 | 0.75 | – | -1.61 | – | -2.36 |
| | PEGASUS | max | – | – | 31.23 | -8.49 | 29.88 | -10.87 |
| | | mean | – | – | – | -2.08 | – | -2.93 |
| | | oracle | – | – | – | -1.15 | – | -1.50 |
| | LSG-BART-base | max | – | – | 30.05 | -6.44 | 26.57 | -8.17 |
| | | mean | – | – | – | -1.77 | – | -2.35 |
| | | oracle | – | – | – | -0.80 | – | -0.99 |
| | GPT-3.5-turbo | max | – | – | 23.86 | -2.46 | 21.68 | -3.92 |
| | | mean | – | – | – | -1.59 | – | -2.76 |
| | | oracle | – | – | – | -0.47 | – | -1.03 |
| WCEP-10 | PRIMERA | max (10) | 0.63 | 0.67 | 35.50 | -1.02 | 48.26 | -0.76 |
| | | mean (9) | 0.66 | 0.64 | – | -0.90 | – | -0.68 |
| | | oracle | 0.67 | 0.67 | – | -0.53 | – | -0.32 |
| | LSG-BART-base | max | – | – | 35.76 | -1.15 | 48.17 | -0.85 |
| | | mean | – | – | – | -1.19 | – | -0.84 |
| | | oracle | – | – | – | -0.88 | – | -0.54 |
| | GPT-3.5-turbo | max | – | – | 26.36 | -0.22 | 32.72 | -0.25 |
| | | mean | – | – | – | -0.06 | – | -0.33 |
| | | oracle | – | – | – | +0.10 | – | +0.11 |
| Multi-XScience | PRIMERA | max (20) | 0.06 | 0.40 | 18.31 | -0.57 | 10.57 | -1.82 |
| | | mean (4) | 0.16 | 0.27 | – | -0.25 | – | -1.27 |
| | | oracle | 0.23 | 0.23 | – | -0.06 | – | -0.97 |
| MS^2 | LED-base | max (25) | 0.16 | 0.22 | 19.66 | -0.14 | 22.74 | -0.47 |
| | | mean (17) | 0.18 | 0.18 | – | -0.10 | – | -0.13 |
| | | oracle | 0.18 | 0.18 | – | -0.01 | – | -0.21 |
| Cochrane | LED-base | max (25) | 0.17 | 0.57 | 17.39 | -0.28 | 23.12 | -2.11 |
| | | mean (9) | 0.31 | 0.44 | – | +0.34 | – | -0.32 |
| | | oracle | 0.40 | 0.40 | – | +0.10 | – | +0.00 |

- **PEGASUS** (Zhang et al., 2020a): PEGASUS is pretrained using a novel Gap Sentences Generation (GSG) objective, where whole sentences from each document are masked, and concatenated to form a pseudo-summary. The model is then fine-tuned on MDS datasets in a supervised fashion.

- **PRIMERA** (Xiao et al., 2022): Extends the GSG objective with a novel masking strategy explicitly designed for multi-document inputs and pre-trains on a corpus of multi-document examples. The model is then fine-tuned on

MDS datasets in a supervised fashion or used in a zero-shot setting.

- **LSG-BART** (Condevaux and Harispe, 2022): Like LED, LSG-BART replaces full self-attention with a sparsified version, dubbed Local-Sparse-Global (LSG) attention, to allow for efficient processing of long inputs. It is initialized with the pretrained parameters of BART and fine-tuned on MDS datasets in a supervised fashion.

Table 10: Results of the open-domain MDS experiments using a dense retriever (Contriever). Difference between a summarizers performance on the ground-truth input documents and performance when the documents were retrieved is shown. Statistically significant results are underlined (paired t-test, p = 0.01).

| Dataset | Model | Top-$k$ Strategy | P@K | R@K | ROUGE-Avg F1 | Δ ROUGE-Avg F1 | BERTScore F1 | Δ BERTScore F1 |
| --- | --- | --- | --- | --- | --- | --- | --- | --- |
| | | | Retrieval | | Summarization | | | |
| Multi-News | PRIMERA | max (10) | 0.21 | 0.80 | 31.66 | -7.77 | 31.78 | -10.47 |
| | | mean (3) | 0.59 | 0.70 | – | -3.31 | – | -4.60 |
| | | oracle | 0.69 | 0.69 | – | -2.20 | – | -3.07 |
| | PEGASUS | max | – | – | 31.23 | -8.69 | 29.88 | -10.88 |
| | | mean | – | – | – | -2.65 | – | -3.45 |
| | | oracle | – | – | – | -1.76 | – | -2.28 |
| | LSG-BART-base | max | – | – | 30.05 | -6.70 | 26.57 | -8.15 |
| | | mean | – | – | – | -2.26 | – | -2.69 |
| | | oracle | – | – | – | -1.41 | – | -1.54 |
| WCEP-10 | PRIMERA | max (10) | 0.63 | 0.67 | 35.50 | +0.10 | 48.26 | +0.90 |
| | | mean (9) | 0.66 | 0.63 | – | -0.14 | – | +0.68 |
| | | oracle | 0.66 | 0.66 | – | +0.29 | – | +0.86 |
| | LSG-BART-base | max | – | – | 35.76 | -0.56 | 48.17 | +0.26 |
| | | mean | – | – | – | -0.96 | – | +0.10 |
| | | oracle | – | – | – | -0.15 | – | +0.66 |
| Multi-XScience[†] | PRIMERA | max (20) | 0.06 | 0.38 | 18.31 | -0.45 | 10.57 | -0.96 |
| | | mean (4) | 0.16 | 0.24 | – | -0.81 | – | -0.96 |
| | | oracle | 0.21 | 0.21 | – | -0.28 | – | -0.37 |
| MS^2 | LED-base | max (25) | 0.18 | 0.25 | 19.66 | -0.43 | 22.74 | -0.70 |
| | | mean (17) | 0.21 | 0.21 | – | -0.37 | – | -0.64 |
| | | oracle | 0.21 | 0.21 | – | -0.32 | – | -0.38 |
| Cochrane | LED-base | max (25) | 0.20 | 0.64 | 17.39 | -0.94 | 23.12 | -2.77 |
| | | mean (9) | 0.35 | 0.49 | – | -0.37 | – | -0.93 |
| | | oracle | 0.44 | 0.44 | – | +0.25 | – | +0.71 |

## C.1 Reproducing Reported Scores

Before experimentation, we attempt to reproduce the reported scores of each MDS model. The results are provided in Table 8. In general, we can reproduce the reported ROUGE scores (and sometimes even improve upon them); however, in a few cases, there are differences as large as ∼3 ROUGE, with the largest differences being observed for PRIMERA, particularly in the zero-shot setting.

## D Evaluation Details

The evaluation metrics, ROUGE and BERTScore, were called from the HuggingFace Evaluate library[23]. Before metrics are calculated, all text is lightly pre-processed by removing leading and trailing whitespace, newline characters and tabs. For ROUGE, we use the default settings besides use_stemmer=True.[24] BERTScore has many parameters which affect the final score; for reproducibility, a hashcode is produced. Our hashcode is: microsoft/deberta-xlarge-mnli_L40_ no-idf_version=0.3.11(hug_trans=4.22.0. dev0)-rescaled_fast-tokenize

---

[23]https://github.com/huggingface/evaluate
[24]https://huggingface.co/spaces/ evaluate-metric/rouge

## E Extended Results from: section 5

In §5, we presented the results from our open-domain MDS experiments for the sparse retriever (BM25) and max top-$k$ strategy only. In Table 9 we present the complete set of results for the sparse retriever. The dense retriever (Contriever) results were comparable and exhibited the same general trends; they are presented in Table 10.

### E.1 Summarization Baselines

In Table 3, we present scores of heuristic baselines. Detailed descriptions of each baseline follow:

- **Random (length-matched) summary**: For each example, take the summary to be the reference summary of *another* example from the same dataset that is the same (or similar) length as the examples reference summary. This provides us with coherent (but likely irrelevant) summaries of approximately the correct length from the same domain.

- **All lead**: For each example, take the summary to be the concatenation of the first sentence from each input document. This is motivated by the notion of a *lead bias*, namely that in many summarization datasets (particularly those comprised of news articles), sentences at the beginning of a document are more likely

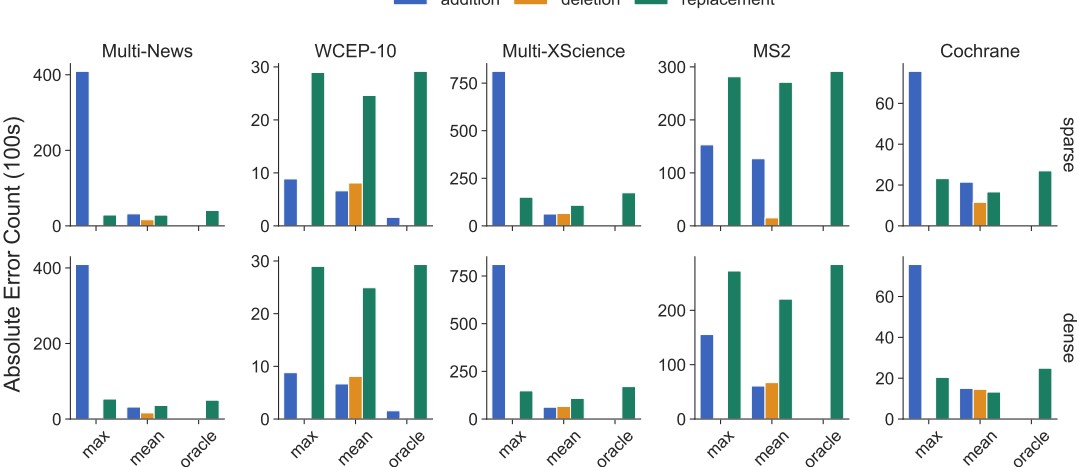

Figure 6: Absolute error counts for different retrieval systems (sparse and dense) and top-$k$ selection strategies (max, mean, oracle). For each example in a given dataset, a retrieved document that does not exist in the ground-truth input document set is counted as an *addition* and a ground-truth document that was not retrieved as a *deletion*. Instances of one addition and one deletion are counted as a *replacement*.

to contain information that appears in the reference summary (Nenkova et al., 2011; Hong and Nenkova, 2014; Xing et al., 2021).

- **Oracle document**: For each example, take the summary to be the input document with the highest ROUGE-1 F1 score with that example's reference summary. This provides us with relevant (but likely incomplete) summaries with high token overlap. A high score may indicate that a dataset is less "multi" (Wolhandler et al., 2022).

- **Oracle lead**: The first sentence of the oracle document (see above). For MSˆ2 & Cochrane, this is the title of the oracle document.

- **Background/abstract**: Applies only to MSˆ2 and Multi-XScience. For each example, take the summary to be the additional input from MSˆ2 (target reviews background section) and Multi-XScience (target articles abstract).

### E.2 Document Retrieval Error Analysis

In Figure 6, we tally the total number of errors made by the sparse (BM25) and dense (Contriever) retrievers on each dataset. For each example, we count an *addition* (i.e. erroneous inclusion) each time a document not in the ground-truth input document set is retrieved, a *deletion* (i.e. erroneous exclusion) each time a ground-truth document is not retrieved and a *replacement* (i.e. an erroneous swap of a relevant document for an irrelevant one)

each time both one addition and one deletion occur. In general, the sparse and dense retrievers make a highly comparable number of errors of each type. Unsurprisingly, the oracle top-k strategy tends to produce the lowest number of errors in total (primarily replacements), followed by mean (a mix of all error types) and max (primarily additions).

## F  Training in the Open-domain Setting

In §6, we presented the results of our experiments fine-tuning summarizers in the open-domain setting. We fine-tuned and evaluated the best-performing model from each dataset: PRIMERA for Multi-News and Multi-XScience, LSG-BART-Base for WCEP-10 and LED-Base for MSˆ2 and Cochrane. Each model was fine-tuned for an additional three epochs (we found additional epochs made little difference) using the original training hyperparameters. All models were fine-tuned with the AdamW optimizer (Loshchilov and Hutter, 2019) in PyTorch via the HuggingFace Transformers library. The learning rate was linearly increased for the first 10% of training steps and linearly decayed to zero afterward. Documents were retrieved using the dense retriever (contriever) and the mean top-k strategy.

## G  Extended Results from: section 7

In §7, we presented results from our experiments simulating document retrieval errors for two model-dataset pairs that exemplified the main trends in the

## Original Document

Relocation of endangered animals carries risks but loss of half of them is highly unusual. Eight out of 14 critically endangered black rhinos have died after being moved to a reserve in southern Kenya, wildlife officials have revealed, in what one conservationist described as "a complete disaster". Preliminary investigations pointed to salt poisoning as the rhinos tried to adapt to saltier water in their new home, the Kenyan Ministry of Tourism and Wildlife said in a statement. It suspended the moving of other rhinos and said the surviving ones were being closely monitored.

## Backtranslated Document

Movement of endangered animals carries risks, but the loss of half of them is very unusual. Eight out of 14 critically endangered black rhinoceros has died after being moved to a reserve in southern Kenya, wildlife officials have revealed in what a conservation expert described as a complete disaster. Preliminary studies pointed to salt poisoning when the rhinoceroses tried to adapt to salt water in their new home, the Kenyan Ministry of Tourism and Animal Health said in a statement. It suspended the movement of other rhinoceros and said that the survivors were being closely monitored.

Figure 7: Graphical depiction of the backtranslation perturbation. A truncated document from the Multi-News (Fabbri et al., 2019) dataset is shown, and changes after backtranslation are highlighted.

Table 11: Results of the sorting perturbation experiments. Difference between a summarizers performance on the ground-truth input documents and performance when the documents were perturbed is shown. Statistically significant results are underlined (paired t-test, p = 0.01).

| Dataset | Model | ROUGE-Avg F1 | BERTScore F1 | $\Delta$ ROUGE-Avg F1 | | $\Delta$ BERTScore F1 | |
| | | | | Random | Oracle | Random | Oracle |
| --- | --- | --- | --- | --- | --- | --- | --- |
| Multi-News | PRIMERA | 31.66 | 31.78 | +0.06 | +0.00 | +0.02 | +0.02 |
| | PEGASUS | 31.23 | 29.88 | -0.05 | +0.04 | -0.05 | +0.16 |
| WCEP-10 | PRIMERA | 35.50 | 48.26 | -0.86 | +0.11 | -0.55 | +0.57 |
| | LSG-BART-base | 35.76 | 48.17 | -0.98 | -0.18 | -0.62 | +0.38 |
| Multi-XScience | PRIMERA | 18.31 | 10.57 | +0.07 | -0.04 | +0.13 | -0.03 |
| MS2 | LED-base | 19.66 | 22.74 | +0.09 | +0.24 | +0.00 | -0.01 |
| Cochrane | LED-base | 17.39 | 23.12 | -0.41 | -0.32 | -0.42 | +0.06 |

rest of the results. We present complete results for all model-dataset pairs in figures 8-15.

### G.1 Backtranslation

In the main paper, we use backtranslation to create token-level perturbations. The procedure involves selecting one or more documents from the input set and translating them to another language and back again, often creating small, token-level changes like paraphrasing and synonym substitution (this is sometimes called "round-trip translation", or RTT). We choose to translate documents to and from Danish, as there exists freely available and high-performing EN→DA and DA→EN machine translation (MT) models. In particular, we use the models provided by the Language Technology Research Group at the University of Helsinki (Tiedemann and Thottingal, 2020). We implement backtranslation using the `nlpaug` library (Ma, 2019). In Figure 7, we provide an example of a backtranslated document demonstrating synonym substitution (e.g. "highly"→"very"), paraphrasing (e.g. "said the surviving ones"→"said that the survivors") and grammatical errors (e.g. "14 critically endangered black rhinoceros has died").

### G.2 Sorting Perturbation Results

In Table 11, we present the tabulated results from the sorting perturbation experiments (see §7 for more details on the experimental procedure and §7.2 for an analysis of the results).

## H Human Evaluation

To make a human evaluation feasible, we chose a single model-dataset pair with high summarization and retrieval performance: PRIMERA and Multi-News.[25] To conduct the evaluation, we randomly sampled 50 examples from the test set and presented three human annotators[26] with the generated summaries for these examples from the "baseline" model (no retrieval) and the open-domain model (with retrieval). Annotators were presented the model summaries in randomized order as "model summary A" and "model summary B" and instructed to select which summary ("A", "B" or "Neither") they preferred for each of two facets,

---

[25]We take the results from the highest performing retriever (sparse) and non-oracle top-$k$ strategy (mean)

[26]The three annotators are a subset of the authors who did not interact with model outputs prior to annotation

Table 12: Examples of degradation of summarization performance in the open-domain setting. Shown is the output of the summarizer in the open-domain setting (truncated) and human annotator comments (paraphrased). The plausible reason for degradation is based on a manual analysis of summarizer inputs and outputs.

| Open-domain model summary | Annotator comments | Plausible reason for degradation |
| --- | --- | --- |
| *For the second year in a row, the Academy of Motion Picture Arts and Sciences did not nominate any black actors to any of the 20 slots in the four acting categories [...] The Hollywood Reporter calls it a "whiteout," and the president of the African American Film Critics Association says [...] There needs to be changes across the board. [...] The Academy, which is 94% white and 77% male, has been trying to diversify its membership [...]* | The **reference** summary indicates that meaningful progress is being made in improving diversity amongst members of the Academy of Motion Picture Arts and Sciences. The **baseline model** gets this correct, but the framing of the **open-domain model** summary is that little to no progress has been made. | The **gold** document set contains 2 documents, each about efforts to improve diversity among members of the Academy of Motion Picture Arts and Sciences, both written *after* the 2016 Oscars. The **retrieved** document set contains 3 documents, 1 from the gold document set, and 2 written *before* the 2016 Oscars, both criticizing the fact that no black actors were nominated for any acting category. |
| *A GoFundMe campaign has raised more than $400,000 for a man who lost his wife in childbirth [...] Christian musician Nathan Johnson gave birth to his first child [...] [his wife] started having complications later in the morning [...] Johnson is surrounded by friends and family who are helping him deal with the loss of his wife [...]* | The **reference** summary is about a GoFundMe campaign created on behalf of Dawn Wells (best known for playing Mary Ann Summers on Gilligan's Island) who is experiencing financial hardship. The **baseline model** gets this correct, but the **open-domain model**'s summary is about a completely different GoFundMe campaign. | The **gold** document set contains 2 documents, both about a GoFundMe campaign for Dawn Wells. The **retrieved** document set contains 1 additional document, the story of a entirely *different* GoFundMe campaign about a musician who lost his wife due to complications during childbirth. |
| *A British woman is "very lucky" to be alive after falling from a cruise ship [...] The cruise line says in a statement that the woman intentionally jumped overboard [...] "the ship and charter company teams are providing support to the family and all impacted guests during this difficult time," [...]* | The summary produced by the **open-domain model** contains an insinuation of possible suicide (i.e. that the woman jumped intentionally) that is not present in the **reference summary** or **baseline model** summary. | The **retrieved** document set contains one additional document compared to the **gold** document set, about a different woman who *did* intentionally jump overboard while on a cruise. |
| *Prime Minister David Cameron said he will step down in two days in favor of Theresa May [...] who will become Britain's second female leader. [...] [S]he will have the task of leading a divided country out of the EU [...] The winner will be announced Sept. 9 and will replace David Cameron [...]* | The **reference** summary indicates that May and Andrea Leadsom are the final two candidates for Prime Minister of the UK, which is reflected in the **baseline model** summary. The summary produced by the **open-domain model** claims both that May is the winner *and* that the winner is yet to be announced. | The **retrieved** document set contains one additional document compared to the **gold** set, about the surprise withdrawal of Leadsom from the race. |

*coverage* and *informativeness*,[27] relative to the provided, human-written target summary $R$:

- **Coverage** (Grusky et al., 2018): How many semantic content units from the reference summary are covered by the model summary.

- **Informativeness** (Nenkova and Passonneau, 2004): How well does the model summary capture the key ideas of the reference summary.

The results from a binomial test[28] on the human annotations, as well as inter-annotator agreement (IAA), are presented in Table 4. Human annotators have a statistically significant preference for the baseline model along both facets, with fair inter-annotator agreement ($\kappa > 0.21$, Landis and Koch, 1977), providing further evidence for the degradation of summarization performance in the open-domain setting observed throughout this work. In Table 12, we provide examples of summaries produced in the open-domain setting alongside (paraphrased) human annotator comments noting issues with the summary. Based on a manual analysis of the inputs and outputs of the summarizer, we also provide plausible reasons for this observed degradation in summarization quality as it relates to the retrieved versus gold document sets.

---

[27]There are many facets for which a human evaluation of summarization could be conducted; we choose coverage and informativeness as rough proxies for recall and precision

[28]https://en.wikipedia.org/wiki/Binomial_test

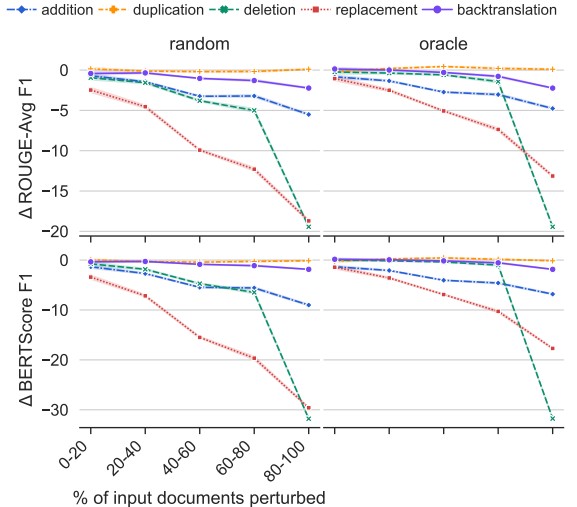

Figure 8: Results of the perturbation experiments on the Multi-News test set using PRIMERA. Mean change in summarization performance plotted against percent of perturbed input documents. 68% confidence intervals (CI) are plotted as error bands.

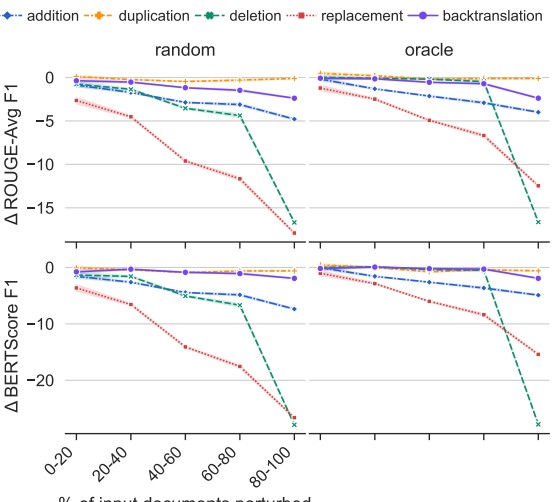

Figure 10: Results of the perturbation experiments on the Multi-News test set using LSG-BART-base. Mean change in summarization performance plotted against percent of perturbed input documents. 68% confidence intervals (CI) are plotted as error bands.

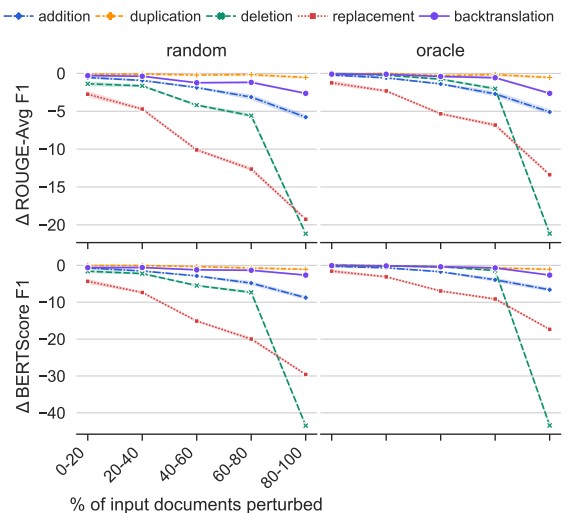

Figure 9: Results of the perturbation experiments on the Multi-News test set using PEGASUS. Mean change in summarization performance plotted against percent of perturbed input documents. 68% confidence intervals (CI) are plotted as error bands.

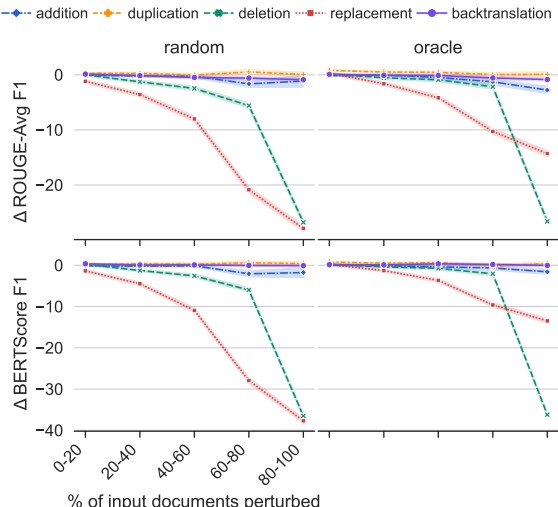

Figure 11: Results of the perturbation experiments on the WCEP-10 test set using PRIMERA. Mean change in summarization performance plotted against percent of perturbed input documents. 68% confidence intervals (CI) are plotted as error bands.

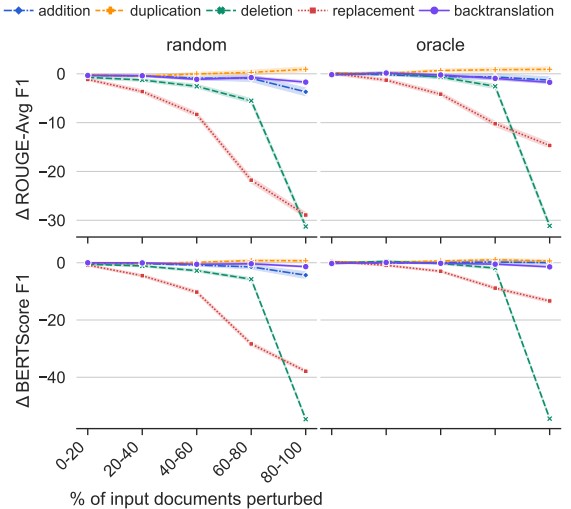

Figure 12: Results of the perturbation experiments on the WCEP-10 test set using LSG-BART-base. Mean change in summarization performance plotted against percent of perturbed input documents. 68% confidence intervals (CI) are plotted as error bands.

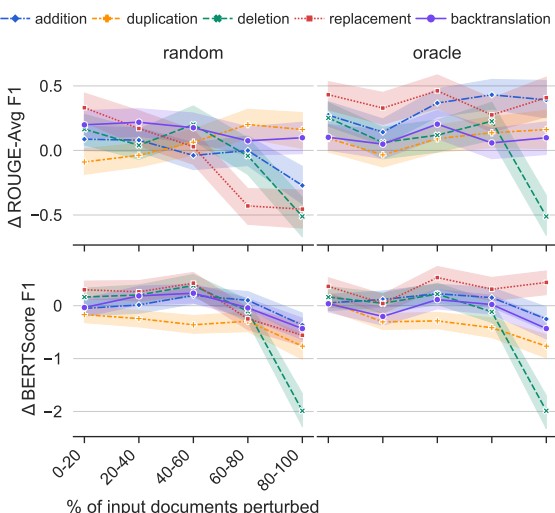

Figure 14: Results of the perturbation experiments on the MS^2 validation set using LED-base. Mean change in summarization performance plotted against percent of perturbed input documents. 68% confidence intervals (CI) are plotted as error bands.

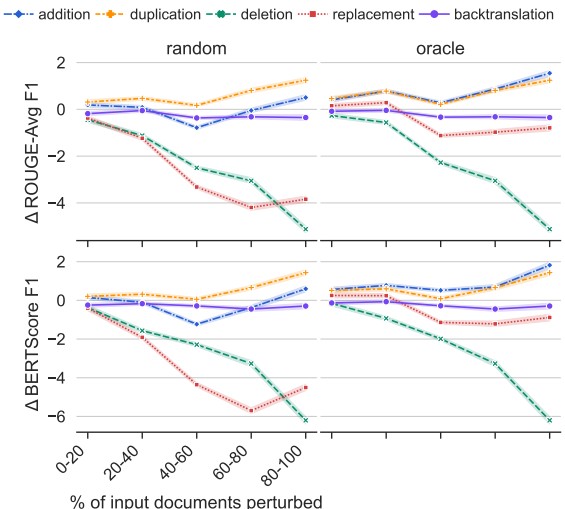

Figure 13: Results of the perturbation experiments on the Multi-XScience test set using PRIMERA. Mean change in summarization performance plotted against percent of perturbed input documents. 68% confidence intervals (CI) are plotted as error bands.

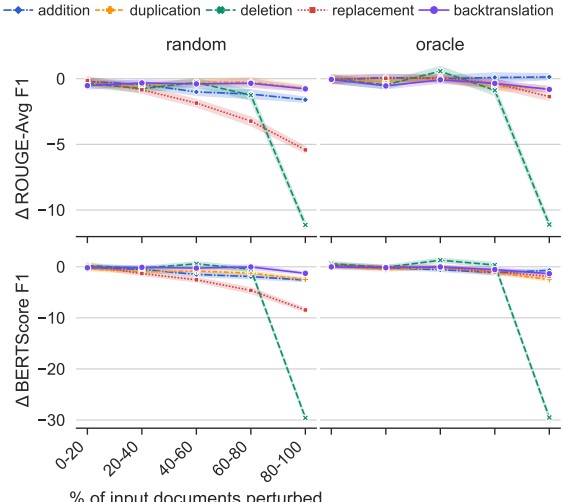

Figure 15: Results of the perturbation experiments on the Cochrane validation set using LED-base. Mean change in summarization performance plotted against percent of perturbed input documents. 68% confidence intervals (CI) are plotted as error bands.