# OpenReview forum: "Open Domain Multi-document Summarization: A Comprehensive Study of Model Brittleness under Retrieval"
_EMNLP/2023/Conference — EMNLP 2023 Findings_

### Official Review · Reviewer_ABNP · 2023-07-31

**Typos Grammar Style And Presentation Improvements:** footnote 19
**Soundness:** 3

**Excitement:**

4: Strong: This paper deepens the understanding of some phenomenon or lowers the barriers to an existing research direction.

**Paper Topic And Main Contributions:**

The work is prompted by a concern that multi-document summarization (MDS) as has been practiced in the literature may not scale well to a real word scenario where one collects documents via information retrieval and conducts summarization. To examine what effect it may have on performance of MDS, the authors simulate the scenario, where summarizers have to work with groups of documents, not curated by humans but collected via conventional retrieval methods. What they found out from the exercise was that MDSs tend to break in the absence of topical coherence among documents.

**Reasons To Accept:**

I will give them some credit for bringing in an interesting perspective for MDS; I like the authors’ breaking down of performance from different angles and discussing how it is influenced by what factors (Figure 4); I also like the way they designed experiments; it is quite meticulous.

**Reasons To Reject:**

It is debatable whether we should put the blame on MDSs for their not being able to cope with topically diverse documents, as they are intended to work under the assumption that documents that come their way are semantically aligned. There is nothing wrong with their breaking down where the assumption does not hold.  There is a chance that the authors are asking a question that people in MDS are not even interested in asking.

**Reproducibility:**

3: Could reproduce the results with some difficulty. The settings of parameters are underspecified or subjectively determined; the training/evaluation data are not widely available.

**Reviewer Confidence:**

4: Quite sure. I tried to check the important points carefully. It's unlikely, though conceivable, that I missed something that should affect my ratings.

---

> ### Author Rebuttal · Authors · 2023-08-28
>
> We thank the reviewer for their succinct and cogent summary of our contributions and for pointing out the meticulousness of our experimental design.
>
> > It is debatable whether we should put the blame on MDSs for their not being able to cope with topically diverse documents, as they are intended to work under the assumption that documents that come their way are semantically aligned. There is nothing wrong with their breaking down where the assumption does not hold.
>
> This is a fair point; we tried to be careful not to suggest this, by noting that the behaviour of existing summarizers in the open-domain setting is _unknown_ (lines 059-061 and elsewhere), motivating our exploratory study. We also fine-tuned the summarizers in this new setting, acknowledging, both implicitly and explicitly, that they should be adapted to the open-domain setting before use (__Section 6__).
>
> However, we understand how the current framing of the paper might muddy this message. We have updated the wording throughout the paper (particularly in the introduction and conclusion) to avoid this ambiguity. The tweaked introduction highlights that our motivation was to study the behaviour of these models in the novel open-domain setting using a _reasonable_ (and even idealized) setup to identify their weaknesses and enable future research on the open-domain setting.
>
> > There is a chance that the authors are asking a question that people in MDS are not even interested in asking.
>
> We hope that the MDS community and broader community of researchers working in summarization will be interested in a task that is more technically challenging (due to the addition of retrieval) and potentially more helpful for an end user with an information need but not necessarily a set of topic-related input documents. Open-domain MDS is an exciting area for future research that can coexist alongside research on traditional MDS tasks without retrieval, much like how open-domain and closed-book QA coexist as research areas.
>
> > footnote 19: the period is missing.
>
> Fixed, thanks.
>
> ---
>
> __RE the reproducibility score__: we are surprised to see a middling reproducibility score. We were extra careful to provide scripts to reproduce all experiments, release all data and model artifacts that we created as part of our study, and we made all the raw experimental results available. We also have notebooks that demonstrate how to reproduce each experiment from the paper. This can be confirmed by checking the anonymous GitHub repository we linked to in our submission. We hope that the reviewer might consider raising the reproducibility score in light of this.

---

### Official Review · Reviewer_GCwU · 2023-08-01

**Soundness:** 3

**Excitement:**

2: Mediocre: This paper makes marginal contributions (vs non-contemporaneous work), so I would rather not see it in the conference.

**Paper Topic And Main Contributions:**

The paper proposes a new dataset for open-domain multi-document summarization and reports a preliminary evaluation of several approaches.
In particular, the authors focus on the impact of retrieval performance on the quality of the final summarization.

**Questions For The Authors:**

Q_A: Please, provide more details about the input size. See the last reject point.
Q_B: Is it possible to combine the ranking of both retrieval models (sparse and dense)? Since they provide different rankings, as stated by the authors, the combination of rankings could lead to better retrieval performance.

**Reasons To Accept:**

- A new dataset for text summarization
- Deep results analysis
- Helpful guidelines for further investigation

**Reasons To Reject:**

- A low novelty for an EMNLP paper
- The paper has nine pages for the main contribution and ten for appendices! This means two things: 1) the paper is hard to read because, at some points, you need to jump from contribution to appendix; 2) the contribution is more suitable as a journal paper than a conference one.
- The authors report the main drawback in the conclusions: "Curating high-quality MDS datasets annotated with queries will be necessary to enable further progress in the open-domain setting.". I think that manually created queries are fundamental for providing a novel and valuable contribution. Using the human-written reference summary as a query biases the retrieval performance since the IR model retrieves documents in which words in the summary occur in the retrieved document, mainly if a BM25 is used as the ranking model.
- Since encoder-decoder architectures accept input of a limited size, the authors should provide a deep evaluation of how this affects the performance. Truncing the document to the first tokens may not be a good choice. You can select the portion of the document that best fits the query.

**Reproducibility:**

4: Could mostly reproduce the results, but there may be some variation because of sample variance or minor variations in their interpretation of the protocol or method.

**Reviewer Confidence:**

4: Quite sure. I tried to check the important points carefully. It's unlikely, though conceivable, that I missed something that should affect my ratings.

**Typos Grammar Style And Presentation Improvements:**

The paper is generally well written but hard to follow since many important content is provided in appendices.
Considering the type of contribution and the content length, in case of rejection, I suggest revising your paper and sending it to a journal.

---

> ### Author Rebuttal · Authors · 2023-08-28
>
> We thank the reviewer for pointing out the depth of our results and analysis and that we produce helpful guidelines for future work on open-domain MDS.
>
> > The paper has nine pages for the main contribution and ten for appendices!
>
> The paper is 8 pages in length (the maximum page limit for EMNLP 2023).
>
> > This means two things: 1) the paper is hard to read because, at some points, you need to jump from contribution to appendix; 2) the contribution is more suitable as a journal paper than a conference one.
> > The paper is generally well written but hard to follow since many important content is provided in appendices.
>
> We appreciate the reviewer's concern, however we were careful to place the most important content for understanding our contributions in the main paper. The appendix consists of experimental details and supporting experiments that are important but not critical to understanding our key contributions. __During a read, there should be little to no jumping between the main paper and the appendix__. We are open to moving some content to the main paper if you have specific suggestions. We will also include as much relevant content in the 9-page allotment provided to camera-ready papers if this manuscript were to be accepted.
>
> > The authors report the main drawback in the conclusions: "Curating high-quality MDS datasets annotated with queries will be necessary to enable further progress in the open-domain setting.". I think that manually created queries are fundamental for providing a novel and valuable contribution. Using the human-written reference summary as a query biases the retrieval performance since the IR model retrieves documents in which words in the summary occur in the retrieved document, mainly if a BM25 is used as the ranking model.
>
> Using the human-written reference summaries as queries was deliberate (__Section 4.4__). Our study aimed to place a lower bound on the degradation of existing summarizers in the open-domain setting. As you say, using reference summaries as queries makes the retrieval problem easier; that we still find large reductions in summarization performance suggests that the degradation would be _worse_ given manually crafted queries. Our perturbation experiments (__Section 7__) generalize our findings, by not relying on specific queries or retrievers and by covering all major types of retrieval errors from 0 to 100% of input documents perturbed.
>
> We also agree that creating a new dataset for MDS with carefully crafted examples and queries would be a major contribution. Our contributions are orthogonal: __we are the first to identify the need for such a dataset and enable future researchers who are better positioned to build it because of our work__.
>
> > Q_A: Please, provide more details about the input size.
>
> The maximum input size of all models is listed in __Table 7__ of the appendix (referenced from __Section 4.3__, which also explains the truncation procedure). For convenience, we relist them below:
>
> - LED: 16,384
> - PEGASUS: 1024
> - PRIMERA/LSG-BART: 4096
>
> Dataset sizes (both in terms of # of docs and # of tokens) are listed in __Table 4__ of the appendix.
>
> > ​​Since encoder-decoder architectures accept input of a limited size, the authors should provide a deep evaluation of how this affects the performance. Truncing the document to the first tokens may not be a good choice.
>
> Our truncation strategy is not unique to open-domain MDS, and the same or similar strategy is used by all baselines we cite.
>
> We note that for 2/5 datasets (MS^2 and Cochrane), no truncation happens at all, as the maximum input size is less than the models’ maximum input size. For a different 2/5 datasets (Multi-News and Multi-XScience), there is no truncation in the _average_ case. For the final dataset (WCEP-10), only a small amount of total tokens are truncated, ~30%, distributed across the ~10 input documents. This can be quickly confirmed by referencing the dataset sizes (__Table 4__) and then models’ max input sizes (__Table 7__). __Given this, we have no reason to suspect that our results would change meaningfully with different truncation strategies__.
>
> > You can select the portion of the document that best fits the query.
>
> This is the difference between _document_-level and _span_-level retrieval. We consider the latter to be beyond the scope of this study, but is certainly an interesting direction. We have already noted this in the __Limitations__ section: “Specialized retrievers may lead to better performance…”
>
> > Is it possible to combine the ranking of both retrieval models (sparse and dense)? Since they provide different rankings, as stated by the authors, the combination of rankings could lead to better retrieval performance.
>
> Yes, a common strategy is to retrieve a large number of documents with a sparse retriever (e.g. BM25) and re-rank those with a dense retriever (e.g. Contriever). Our preliminary results showed marginal gains in retrieval performance (+0.05 P/R@K at best) with this pipeline. __Because of the marginal gains and because it complicates the relationship between retriever type and impact on summarization performance, we elected not to include these experiments in the paper__. We have updated the text in the __Limitations__ section (“Specialized retrievers may lead to better performance…”) to note this.
>
> ---
>
> __RE the soundness score__: we are surprised to see a low soundness score. We were extensive and meticulous in our experimental design; something the other reviewers noted. Alongside our main experiments evaluating and training models in the open-domain setting, we included an extensive set of perturbations designed to mimic all main types of retrieval errors, from 0-100% of input documents modified. Finally, the appendix contains supplementary experiments that support our main claims. We were also careful to ensure everything was reproducible by including notebooks demonstrating how to run each experiment from the paper (this can be confirmed by checking the anonymous GitHub repository we linked to in our submission). We hope that the reviewer might consider raising the soundness score in light of this.

---

### Official Review · Reviewer_7Smv · 2023-08-03

**Soundness:** 4

**Excitement:**

4: Strong: This paper deepens the understanding of some phenomenon or lowers the barriers to an existing research direction.

**Paper Topic And Main Contributions:**

The paper establish the new task of open domain MDS. It creates a new dataset, shows that current MDS models perform lower on this task, while fine-tuning improves the results. The paper also makes a nice error analysis.

**Questions For The Authors:**

line 268: "lower-bound" -should it be "higher bound" or am I missing something?

**Reasons To Accept:**

- The paper is well written and the flow is clear.
- The paper points out a real lack in the current MDS setup which does not align with real-world applications.
- The authors conducted extensive experiments, deriving several insights.
- The authors added a thorough error analysis.

**Reasons To Reject:**

- The authors claim that the data is more difficult than other dataset as it contains many irrelevant documents that could hurt summarization. On the other hand, in a real-world setup there might be many "difficult" documents, as irrelevant documents that seem to be relevant as they are similar to relevant documents. Adding analysis quantifying this phenomenon in the new dataset may add additional insights.
- In fact, the paper contribution focuses on pointing out the need for a new setup. In practice, the suggested dataset can be used as an upper bound only. A significant contribution could be by releasing a "real" dataset. But it might be okay to leave it for future work.

**Reproducibility:**

4: Could mostly reproduce the results, but there may be some variation because of sample variance or minor variations in their interpretation of the protocol or method.

**Reviewer Confidence:**

4: Quite sure. I tried to check the important points carefully. It's unlikely, though conceivable, that I missed something that should affect my ratings.

---

> ### Author Rebuttal · Authors · 2023-08-28
>
> We thank the reviewer for commenting that our paper is well written, points out a gap between current MDS research and real-world applications, and that our experimentation is extensive. We address any questions or concerns below:
>
> > The authors claim that the data is more difficult than other dataset as it contains many irrelevant documents that could hurt summarization. On the other hand, in a real-world setup there might be many "difficult" documents, as irrelevant documents that seem to be relevant as they are similar to relevant documents. Adding analysis quantifying this phenomenon in the new dataset may add additional insights.
>
> We agree; we carefully designed experiments to analyze this phenomenon in __Section 7__. In the _oracle_ setting of the _addition_ perturbation, irrelevant documents were selected based on their similarity to the target reference summary (see __Section 7.1__). We believe this is exactly the experiment you are suggesting.
>
> In short, we found that the performance degradation tends to be _similar_ or _slightly greater_ in the random setting. A possible explanation is that random additions introduce more _topically_ and _semantically_ irrelevant content than oracle additions and that the automatic metrics may be particularly sensitive to this content when it is included in the model’s output summary (e.g. names, places and events not present in the ground-truth summary). We have updated the main text to note this relationship and include the provided rationale.
>
> > In fact, the paper contribution focuses on pointing out the need for a new setup. In practice, the suggested dataset can be used as an upper bound only. A significant contribution could be by releasing a "real" dataset. But it might be okay to leave it for future work.
>
> We agree that a new dataset for open-domain MDS would be an exciting contribution, however, our contributions are orthogonal. __By identifying the problem, confirming that existing summarizers are sensitive to retrieval, and providing a thorough set of guidelines, we have enabled future researchers to collect, curate and create a “real” dataset.__
>
> > line 268: "lower-bound" -should it be "higher bound" or am I missing something?
>
> We consider our results a _lower_-bound on the expected performance _degradation_, not an _upper_ bound on performance. Several aspects of our experimental design are intentionally idealized, and performance degradation of summarization in the true setting is likely even worse, hence the _lower_ bound. Please see __Section 4.4__ where this is explained in detail.

---

### Meta-Review · Area_Chair_aKNg · 2023-09-11

**Recommendation:** 4

**Metareview:**

This paper investigates how multi-document summarization (MDS) models behave when the set of input documents is provided by an IR model. It formalizes this task as open-domain MDS, bootstraps it using existing datasets and summarization/retrieval models, and conducts extensive experiments. Reviewers agree that the paper makes meaningful contributions (recasting MDS in a more "realistic scenario", thorough experiments and analysis, opening potentially enabling further work in MDS) and I believe some of their concerns were addressed in the rebuttal period.

---

### Decision · Program_Chairs · 2023-10-07

**Decision:**

Accept-Findings

**Comment:**

This paper investigates how multi-document summarization (MDS) models behave when the set of input documents is provided by an IR model. It formalizes this task as open-domain MDS, bootstraps it using existing datasets and summarization/retrieval models, and conducts extensive experiments. Reviewers agree that the paper makes meaningful contributions (recasting MDS in a more "realistic scenario", thorough experiments and analysis, opening potentially enabling further work in MDS) and I believe some of their concerns were addressed in the rebuttal period.